# Experimental Study on Vibration Attenuation Characteristics of Ballastless Track Structures in Urban Rail Transit

**DOI:** 10.3390/s25030868

**Published:** 2025-01-31

**Authors:** Wuji Guo, Zhiping Zeng, Mengxuan Ye, Fushan Liu, Weidong Wang, Cheng Chang, Qiuyi Li, Ping Li

**Affiliations:** 1School of Civil Engineering, Central South University, Changsha 410083, China; guowujisx@gmail.com (W.G.);; 2MOE Key Laboratory of Engineering Structures of Heavy Haul Railway, Central South University, Changsha 410075, China; 3Department of Structural Engineering and Building Materials, Ghent University, 9000 Ghent, Belgium; 4School of Engineering, Monash University Malaysia, Subang Jaya 47500, Malaysia; 5China Railway Fourth Survey and Design Institute Group Co., Ltd., Wuhan 430063, China; 6Guangzhou Metro Design and Research Institute Co., Ltd., Guangzhou 510010, China

**Keywords:** hammering rail experiment, long sleeper integrated slab, vibration attenuation law, time–frequency analysis

## Abstract

With the rapid development of urban rail transit, the intensity and impact range of train-induced vibrations are increasing. Investigating the transmission characteristics and attenuation patterns of these vibrations in track structures aids in understanding train-induced environmental vibrations. This study conducted rail impact experiments on a long sleeper integrated slab of a straight section of a subway tunnel. The hammer struck the rail at various positions, and acceleration sensors recorded the responses of the rail, slab, and tunnel. In order to determine the impact force, the vertical wheel–rail force and the vibration response of track structures were measured. Then, the Lance-LC1304B force hammer was selected for the experiment, and the hammer impact force reached 30 kN, the magnitude of which reached the measured wheel–rail force size for the line. Based on the results of the impact tests, the vibration attenuation characteristics of the track structure were analyzed. Accordingly, reference values for the truncation time and truncation distance in the vehicle–track coupled dynamics model’s moving window were provided. By comparing the results of the hammering experiment with the train-induced vibration results, the main excitation frequencies during train operation were determined. These findings provide valuable insights for the development of rail transit systems.

## 1. Introduction

With the rapid development of urban rail transit, the rail network has become increasingly dense and widespread. While facilitating urban residents’ commuting, the vibrations induced by trains have also brought negative impacts to vibration-sensitive areas such as hospitals, schools, historical buildings, and residential areas [1,2,3,4,5,6,7,8].

Conducting vibration experiments on track structures can authentically showcase the characteristics and attenuation patterns of rail substructure vibrations. On the one hand, this can provide data support for predicting rail substructure vibrations. On the other hand, it can also serve as a basis for the design optimization of vibration reduction measures.

Over the past 20 years, numerous scholars have conducted experiments on the vibration of track structures in laboratory. Zeng et al. [9] analyzed the vibration characteristics of concrete sleepers and rubber composite sleepers through drop harmmer experiments, investigating the vibration control effects of rubber composite sleepers. Wang et al. [10] conducted drop wheelset experiments on the CTRS-type track slab. They compared and discussed in detail the vibration characteristics, dynamic stiffness, and damping coefficients of different ballastless track systems under different wheel drop heights. Zeng et al. conducted a drop wheelset experiment to study the vibration reduction effects of a new LVT track structure used in heavy haul railways [11]. They also investigated the impact of temperature environments on the vibration characteristics of CRTS I double-block ballastless track [12]. Huang et al. conducted drop wheelset experiments on both steel spring floating slabs [13] and rubber damping pad floating slabs [14] to assess their vibration reduction effects. By comparing the ground vibration responses under drop wheelset excitation before and after fatigue experiments, they analyzed the impact of fatigue loads on the vibration reduction effects of these two mitigation measures. Although these experiments have revealed certain vibration characteristics of track structures, the laboratory settings differ from actual operational environments, potentially leading to deviations in results. Moreover, the conclusions of these studies predominantly focus on the frequency characteristics, vibration reduction effects, and vibration transmission within track structures, with limited attention to the duration and attenuation distance of vibrations.

Field experiments on the vibration characteristics of track structures are relatively scarce. T. Jaquet et al. [1] conducted tests on the ground vibration response caused by train operations in sections with floating slabs installed along Beijing Subway Line 8. They analyzed the necessity of installing steel spring floating slabs in sensitive areas. Ma et al. [15], using the Xi’an Bell Tower as an example, explored the effectiveness of using steel spring floating slabs to control ground vibrations of historical buildings through field tests and theoretical model analysis. Qu et al. [16] conducted field experiments in Shenzhen, China, measuring the accelerations of rail and ground in straight sections of double-line subway, and compared the differences in ground vibration caused by trains running on near-line versus far-line. Cao et al. [17] performed vibration tests on a seven-story residential building constructed above an elevated depot, measuring acceleration time histories at different locations and comparing their respective spectra. Liang et al. [18] conducted detailed field measurements on the sources and transmission characteristics of subway vehicle-induced vibrations, and they found that the vibrations induced by trains on the test tracks and throat of frogs were greater than in other areas. These experiments aimed either at evaluating vibration mitigation effects or verifying the accuracy of predictive models. Studies on vibration attenuation mostly focus on the decay rate along the rail, with few addressing the attenuation characteristics of track slabs, tunnels, and other structures.

Due to the continuous excitation caused by the train load moving longitudinally along the rail during train operation, it is challenging to accurately quantify the excitation position, excitation time, and magnitude of the train load. Therefore, it is difficult to study the duration and attenuation distance of vibrations in track structures directly through train excitation. Hammering experiments address these issues by allowing precise control of the excitation position, time, and magnitude. Therefore, this article conducted hammering tests on actual subway lines.

Currently, the analysis focus of hammering experiments conducted on track structures within tunnels is primarily on the vibration attenuation rate of the rail [19] or on the modal analysis of certain track structures [20]. However, there are few studies on the vibration attenuation characteristics of track structures. Moreover, the existing standards for track hammering experiments [21,22] involve relatively small hammering forces, which cannot generate sufficient energy to excite vibrations in the track slab and tunnel. Therefore, they are inadequate for studying attenuation patterns.

In vehicle–track coupling dynamic models, it is often necessary to establish long track models. However, longer track models lead to a significant increase in degrees of freedom, making it very slow to directly solve the entire model. In practice, when calculating the vibration response of track structures, it is sufficient to consider only the sections of the entire track that are actively vibrating at a given time step, rather than computing the entire model. This approach significantly improves the efficiency of model resolution. This necessitates providing the vibration decay time and decay distance, which correspond to determining the truncation time and distance in the moving window method. In existing studies, researchers such as Dong [23] have proposed “cutting and connecting” for track modeling, while Koh et al. [24] introduced the moving element method, and Song S. [25] established the sliding window method, all of which address the issue of determining truncation time and distance. Most existing research has relied on numerical experiments to establish values for these two key parameters, with no experimental studies reported on parameter selection. The parameters determined through numerical experiments can be significantly influenced by model parameters, numerical algorithms, and other factors, leading to potential discrepancies with actual conditions. Therefore, this paper conducts a detailed investigation of these two key parameters based on experimental data and provides values for decay distance and decay time derived from the experiments, thereby filling the gap in experimental research on these critical parameters.

To fill the gaps in the current research, it is necessary to design and conduct an experiment on actual track structure inside the tunnel. This experiment should clarify the frequency characteristics of track structure vibrations, their duration, their attenuation distance, and the transmission characteristics of vibrations between track structures. This paper conducts a rail hammering experiment within a shield tunnel laid with long-sleeper ballastless track slabs. Using a force hammer to strike the rail, acceleration sensors record the responses of the rail, slab, and tunnel at various hammering locations. By analyzing the experimental results in the time domain, frequency domain, and time–frequency domain, the vibration duration, the spatial attenuation laws of vibrations at different frequencies on various track structures, and the frequency changes and dissipation of vibrations as they transmit between structures were revealed.

Based on the conclusions related to the duration of the vibration response and attenuation distance, reference values for the truncation time and truncation distance in the moving window method of train–track coupled dynamics are provided. Additionally, the structural response of the experimental section during actual train operation was tested. By combining the analysis with the hammering experiment, the main excitation frequencies of the train were obtained.

The findings regarding the frequency characteristics of vibrations and their transmission characteristics between structures showcase the inherent properties of the structures. These conclusions are beneficial for the design of vibration mitigation measures, allowing for the development of isolation facilities tailored to the vibrations of different frequencies in different structures.

## 2. Experimental Methods

Measuring the vibration response of the track structure directly under train excitation is the most effective approach to studying the attenuation characteristics. While we can measure the magnitude of the wheel–rail forces, it is important to note that the timing of the wheel–rail force excitation is difficult to control. During measurements, there may be instances where the vibration caused by the excitation from one wheel has not yet attenuated, while the excitation from another wheel has already begun to act. This phenomenon of overlap is particularly evident in the vibration responses of the track structure recorded in Section 5. In the presence of such uncontrolled overlapping signals, investigating the vibration attenuation characteristics of the track structure becomes considerably challenging.

To address this issue, this paper employed a force hammer to provide excitation, allowing for complete control over the excitation process. This approach ensures that the test signal is not influenced by other excitations before it has fully attenuated.

For convenience, this paper stipulates the following conventions: the origin is the intersection point of the wheel–rail contact surface, the correspondence to the line at the rear end of the train, and the longitudinal symmetry plane of the train. The positive direction of the *X*-axis is along the direction of train travel, the positive direction of the *Y*-axis is to the right of the train travel direction, and the downward direction is the positive direction of the *Z*-axis.

Before conducting the experiments, it is essential to determine the magnitude of the hammer impact force to ensure that it closely resembles the actual wheel–rail force. To achieve this, we measured the wheel–rail force and the vibration response of the track structure under operating conditions. The wheel–rail force can be measured either directly by monitoring the stress changes in the wheelset [26,27,28] or indirectly by measuring the stress changes in the rail [29,30]. However, during the passage of the train through the experimental section, measuring the stress in the wheelset does not provide an accurate representation of the wheel–rail force acting on the section. Therefore, this study employed the indirect method of measuring the stress in the rail. In this method, the strain gauge rosettes are affixed to the rails to measure the vertical forces exerted on the rail by the train.

Figure 1 illustrates the placement and bridging method of the strain gauge rosettes used for measuring vertical wheel–rail forces. Each measurement point requires the installation of four strain gauge rosettes, adhered to both sides of the rail web. The placement on each side is symmetrically distributed around the center of the fastener spacing, with a spacing *b* of 240 mm. The strain gauge rosettes are oriented at a 45° angle to the longitudinal axis of the rail.

After the strain gauge rosettes are installed, it is necessary to calibrate the relationship between force and strain. Figure 2 illustrates the calibration setup for vertical wheel–rail forces. The setup involves using a hydraulic jack in conjunction with a reaction frame to apply lateral and vertical loads to the designated rail positions, recording the corresponding relationship between the applied load and the output of the strain gauge. For vertical force calibration, loads are incrementally increased to 100 kN in increments of 10 kN. The average of three valid calibration readings is taken as the calibration value.

In the experiment, we used the strain gauge rosettes model BE120-3BA and recorded the strain of the rails using the DH5981 type dynamic data acquisition instrument, with a sampling frequency set to 100 Hz.

In addition to investigating the time-domain decay characteristics of vibrations, this study also aims to explore the spatial decay patterns of track structure vibrations. Therefore, after determining the magnitude of the impact force, it is necessary to select the hammer strike location. We referenced the hammer strike point selection methods used in measuring rail vibration decay rates from the literature [19] and standards [21,22]. Unlike the testing of rail vibration decay rates, the hammer strike points in this study are uniformly arranged at mid-span positions. This configuration ensures that the strike points are located at the same position within the periodic structure of the rail, resulting in a more systematic decay of the vibration signals. It also helps to avoid the influence of striking at the nodes of the modal shapes on the vibration decay characteristics.

The selection of the hammering positions is depicted in Figure 3. The red numbers above represent the hammering position indexes, while the black numbers below indicate the interval indexes between fasteners. The test acceleration section is designated as the 0th fastener interval. Therefore, the 1st hammering position is located on the rail head surface of the rail at the 0th fastener interval position. Moving along the direction of train travel, the 2nd position is located one fastener interval away from the 1st fastener interval position. As the distance from the test section increases, the spacing between hammering positions gradually increases.

In dynamic simulation calculations, the modal superposition method is commonly employed to solve the dynamic response of rails or other substructure components. The solution formula is typically expressed as:(1)$$X_{r}\left(x,t\right)=\sum_{k=1}^{N_{r}}Z_{k}\left(x\right)q_{k}\left(t\right),\;\left(k=1,2,\ldots ,N_{r}\right)$$
where $X_{r}\left(x,t\right)$ represents the response of the structure at position *x* and time *t*, $Z_{k}\left(x\right)$ denotes the *k*-th mode shape function of the structure in generalized coordinates, and $q_{k}\left(t\right)$ refers to the generalized modal coordinate. According to modal theory, the excitation position can theoretically influence experimental results. When the excitation occurs at a modal node, the frequency response corresponding to that mode may be absent. However, for the track structure, which is a periodic system, all hammering points in the experiment were located at rail positions between fasteners. Theoretically, these hammering points are situated at equivalent positions within the local modal shapes of the periodic structure. This ensures that the modal frequencies excited by the hammering points discussed in this paper remain consistent.

The experiment involved hammering the surface of the rail head with a force hammer to apply pulse loads to the track structures. The impact response of the track structures was measured by the vibration acceleration of the rail, slab, and tunnel.

This section designs an experimental plan for the subway laid with long sleepers and integral slabs. Acceleration measurement points were placed at the midpoint of the rail clip on the slab, as illustrated in Figure 4. The acceleration measurement points for the rail, slab, and tunnel were positioned along the same cross-section. The rail measurement point was directly above the rail head, the slab measurement point was located 10 cm from the edge of the slab, and the tunnel measurement point was positioned 1.25 m from the rail surface.

The hammer used in the experiment, as shown in Figure 5, is Lance-LC1304B, equipped with a force sensor with a range of 300 kN, and it features a steel head. The sensor used to measure the response of the rail, as shown in Figure 6, is a DH-1A001 accelerometer with a range of 1000 g. The sensor used to measure the response of the slab, as shown in Figure 7, is a CF0162 accelerometer with a range of 25 g. The sensor used to measure the response of the tunnel, as shown in Figure 8, is an LC-0134M accelerometer with a range of 2 g. The responses of the track structures under impact loads were recorded using a DH5981 type dynamic data acquisition instrument, with a sampling frequency set to 20 kHz.

After confirming the good construction and installation status of the track structures, as well as the normal operation of the acceleration sensors and data acquisition equipment, impact loads are sequentially applied at the load application positions. The impact force and the response acceleration of the track structures under the impact load are recorded for each loading. Five loads are applied at each position, ensuring that the hammering loads are essentially consistent.

Due to the complex conditions of the tunnel environment during field experiments, there might be other excitation sources aside from the hammering force, such as road vehicles traveling above the tunnel, nearby construction activities, or passing trains in adjacent tunnels. These background noises could influence the analysis of experimental results. To address this issue, the experiments were conducted between 2:00 a.m. and 4:00 a.m., a time chosen to avoid the majority of these external vibration sources. Furthermore, to further reduce the impact of background noise, five valid hammer strikes were performed for each hammering point during the experiments. In post-processing, the hammering signals were averaged over these five trials, which not only minimized the influence of random noise but also ensured the reliability of the experimental results.

It is important to note that although the experimental method in this study is similar to that used for testing rail decay rates and modal responses, there are fundamental differences in terms of hammer force, measurement points, and analysis objectives.

Although the selection of hammer impact points is similar to that described in the literature [19] and standards [21,22] regarding rail decay rate testing, there are essential differences between the two experiments. The decay rate tests aim to measure the spatial decay rate of vibrational energy as it propagates along the rail, without considering the temporal decay of vibrations. Moreover, the analysis is limited to the vibrational energy of the rail itself, overlooking the differences in how vibrations of varying frequencies decay with distance. Additionally, the hammer force used in those tests is significantly smaller than the actual wheel–rail interaction force. In contrast, this study employs a hammer force that is comparable to the actual wheel–rail force, analyzing the decay patterns of vibrations in the rail, track slab, and tunnel across time, space, different structures, and various frequencies. This approach not only enriches the scope of the research but also takes into account the decay characteristics of vibrations across multiple dimensions.

Similarly, this study differs from conventional modal testing that employs hammer impacts. While both utilize a force hammer for excitation, the hammer force in modal testing is determined to optimally excite the common modes of the structure, with hammer points chosen to avoid nodal locations. The primary goal of modal testing is to identify the inherent frequencies and modal shapes of the track structure. In contrast, the hammer force used in this study is specifically selected to simulate the magnitude of the wheel–rail force, and the selection of hammer points is tailored to serve the experimental objective of analyzing the spatial decay patterns of vibrations.

In addition, the analytical results of this study can provide experimental data support for determining the truncation time and distance in the moving window of numerical simulations, addressing a gap in this field that has not been covered by other experiments.

## 3. Section Condition

The experimental site is depicted in Figure 9. The experiment was conducted on a subway line in Changsha, China. The experimental section was located inside a circular shield tunnel, with a diameter of 5.40 m and a depth of approximately 16.70 m. The height of the rail structure is 0.78 m.

This section installed with DZ III fasteners. The vertical and lateral stiffness of the fasteners are 5 $\times \;10^{7}N/m$ and 3 $\times \;10^{7}N/m$, respectively. The structures of the DZ III fastener is shown in Figure 10.

This section features a cast-in-place continuous rail-sleeper integral ballastless slab shown in Figure 11. The pre-stressed concrete sleepers are 2.10 m in length, spaced 0.60 m apart, and made of C60 concrete. The slabs are 2.37 m wide, 0.30 m high, and 11.97 m long, with a 0.03 m gap between slabs, and are made of C35 concrete.

The soil layers near the experimental section consist of various materials, including miscellaneous fill, pure fill, silt, clayey silt, loess, fine sand, rounded gravel, cobblestone, residual clayey silt, strongly weathered argillaceous siltstone, and moderately weathered argillaceous siltstone.

## 4. Analytical Method

The workflow of this study is illustrated in Figure 12. To reveal the characteristics of vibration transmission in the track structures, it is necessary to process and analyze the experimental results. The vibration signals recorded by sensors are analyzed in the time domain, frequency domain, and time–frequency domain. In this section, we will briefly introduce the signal processing methods used in the analysis process.

During time-domain analysis, on the one hand, by directly comparing the variation pattern of peak values of vibration acceleration time history curves over time, the attenuation of the vibration amplitude over time is analyzed. On the other hand, by comparing the effect of changes in hammering positions on the vibration at measurement points, the attenuation of vibration amplitude with distance is analyzed.

The Frequency Response Function (FRF) is a mathematical representation of the relationship between input and output of a system in the frequency domain. It is widely used in fields such as structural dynamics, control systems, and signal processing to characterize the dynamic behavior of systems. By analyzing the FRF, one can understand the resonant frequency, damping characteristics, and frequency-dependent behavior of a system.

When conducting frequency domain analysis, following the method described in reference [31], the estimate of the vibration acceleration FRF, denoted as $H_{1}$, is recorded as follows:(2)$$H_{1}\left(f\right)=\frac{X\left(f\right)F^{*}\left(f\right)}{F\left(f\right)F^{*}\left(f\right)}$$
where F(f) and X(f), respectively, represent the spectra of the input signal and the output signal. In this context, the input signal is the force hammer excitation signal, and the output signal is the acceleration response of the track structures. The symbol ∗ denotes the complex conjugate of the signal.

To reduce the influence of noise, it is necessary to average the frequency response functions obtained from the five hammer strikes. The peaks of the FRF reflect the frequency characteristics of the track structures. By extracting the magnitude of the FRF at resonance peak frequencies from responses at different hammering positions, one can obtain the attenuation law of track structures vibration frequency with distance.

Time–frequency analysis is a method that jointly analyzes signals in the time–frequency domain. It combines the advantages of time-domain analysis and frequency-domain analysis, revealing the characteristics of signal variations over time and frequency. Through time–frequency analysis, changes in signals over different time periods and frequency ranges can be observed, providing a more comprehensive understanding of the dynamic properties of signals and the distribution patterns of frequency components. Common time–frequency analysis methods include Short-Time Fourier Transform (STFT) and Continuous Wavelet Transform (CWT).

Through time–frequency analysis, it is possible to discuss the attenuation characteristics of vibrations at different frequencies over time. Due to the advantage of having an adaptive window, this paper employs the Morlet wavelet as the basis function for continuous wavelet transform to process the vibration signals.

## 5. Structure Response of a Passing Train

In order to clarify the required hammering force during the experiment, we first tested the wheel rail force and vibration response of the track structure on the experimental section under actual driving conditions. This line adopts a six-car formation of B-type metros, and the running speed of the train on the test section is about 60 km/h.

During the calibration of wheel–rail forces, the relationship between load and strain was obtained. By fitting the data, the mapping relationship between loads (both inner and outer rails) at the two wheel–rail force measurement points and strain was derived, as shown in Figure 13a,b. The R-squared values of the fitted equations for the four measurement points are 1.00 and 0.99, respectively, indicating a high degree of fit. This ensures that the strain data recorded when the wheelsets pass can be accurately converted into wheel–rail forces.

Using the calibration equations derived in Figure 13a,b, the strain measurements taken during train operation were converted into wheel–rail forces.

The time history curves of vertical wheel–rail forces at the measurement point are shown in Figure 13c. In these curves, the vertical wheel–rail force is positive in the downward direction. The figure clearly distinguishes the loads generated as each wheelset passes the measurement point.

From the figure, it can be observed that the vertical forces on the inner and outer rails are nearly identical in the straight-line section. When the wheels pass over, the peak vertical wheel–rail forces range from 27.87 kN to 40.06 kN.

Figure 14 presents the time history vibration responses of the rail, slab, and tunnel induced by train. From the time-domain curves, it can be observed that the vibrations of the rail, slab, and tunnel exhibit a spindle-shaped waveform. The acceleration peak of the rail reaches 57.22 g. The slab vibration is significantly smaller than the rail acceleration, with a maximum peak value of only 0.31 g. The tunnel vibration is slightly less than the slab, with a maximum acceleration peak value of 0.15 g.

From Figure 13c and Figure 14, it is evident that although the wheel–rail force excitation is discrete and clear, the vibration responses of the rail, slab, and tunnel exhibit overlapping excitations from different wheelsets. Consequently, directly analyzing the vibration responses of the track structure during train operation makes it challenging to determine the attenuation time and distance, as well as other characteristics of vibration decay for each structure.

Figure 15 shows the Power Spectral Density (PSD) and time–frequency characteristic diagrams of the rail, slab, and tunnel. The PSDs of the rail, slab, and tunnel all show peak values below 1400 Hz, occurring at approximately 70 Hz, 200 Hz, 400 Hz, 600 Hz, 900 Hz, and 1100 Hz. From Section 6, we see that under pulse excitation, the resonance peaks of the rail also appear around 200 Hz and 1100 Hz.

Additionally, there is a resonance peak around 70 Hz in the slab. Since the slab remains in a continuous vibration state as the train passes, it can be inferred that the peak values of the rail and tunnel PSD around 70 Hz under train excitation are forced vibration responses, originating from the slab’s vibration around 70 Hz. In summary, the peaks at 400 Hz, 600 Hz, and 900 Hz are primarily caused by the vehicle’s own vibrations and are less influenced by the track structure.

In the rail’s time–frequency characteristics plot, significant distinctions can be observed between 50 Hz and 150 Hz, corresponding to the passing moments of each wheel. In the slab’s time–frequency characteristic diagram, there is noticeable noise interference around 50 Hz, and its vibration amplitude does not change over time. The individual wheelsets can still be distinguished within the frequency range of 50 Hz to 150 Hz, and the vibration amplitude at each frequency is significantly attenuated compared to the rail. In the tunnel’s time–frequency characteristic diagram, individual wheelset pass-by moments can no longer be distinguished.

## 6. Results of the Rail Hammering Experiment

### 6.1. Force Hammer Signal

Based on the measured wheel–rail forces during train operation, this study set the hammering force at 30 kN. Regarding the frequency analysis range, it is determined through processing of the force hammer signal. The time-domain waveform and power spectral density (PSD) curve of the load signal applied to the rail by the force hammer are depicted in Figure 16.

Compared to the vibrations induced by train, the responses under hammer excitation are smaller, as the vibrations from the train are the result of the superposition of multiple wheel-pair excitations. Thus, it can be considered that the vibrations generated by the hammering are comparable to those during train operation.

From the time history of the hammering signal shown in Figure 16a, it can be observed that the load approximates an impulse signal, with the magnitude controlled at around 30 kN. Comparing this with the response of the track structure under actual train operation conditions, as tested in Section 5, the peak vibration acceleration responses of the rail, slab, and tunnel under this hammering force reach 56.80%, 64.51%, and 66.67% of the actual operational conditions, respectively. This indicates that such hammering is sufficient to excite the modal participation factors of higher modes in the track structure.

From Figure 16b, it can be seen that the excitation frequency range remains stable below 1200 Hz and rapidly attenuates beyond 1200 Hz. It can be inferred that this excitation can sufficiently stimulate the response of the system below 1200 Hz [20]. Based on the coherence between the excitation signal and the acceleration response, a cutoff frequency of 1200 Hz is chosen for the analysis.

### 6.2. Rail Vibration Characteristics

Figure 17 depicts the acceleration response of the rail when subjected to impact loads at different positions. Due to the varying distances between hammering points, the acceleration response corresponding to each hammering position is plotted against the distance from the hammering point to the test location. In the figure, each curve represents the response from a different hammering position. From the graph, it can be observed that when the hammering point is located at the testing cross-section, the rail exhibits the maximum vibration response, with a maximum acceleration of approximately 32.50 g. As the hammering position gradually moves away from the acceleration measurement point, the initial response time of the rail gradually lags behind, and the maximum vibration acceleration of the rail also decreases accordingly. The vibration of the rail gradually attenuates over time, with its response essentially ending after approximately 0.02 s.

The $H_{1}$ estimate of the rail vibration frequency response function (FRF) is shown in Figure 18. From the graph, it can be observed that the first-order resonance peak of the rail FRF appears at 195 Hz. The generation of this resonance peak is mainly influenced by the characteristics of the rubber pad in fastener system, and its magnitude and frequency vary with changes in the mechanical properties of the pad [32,33]. The second-order resonance peak appears at 1028 Hz, which is generated by the pinned-pinned resonance of the rail [32,33].

The variation of resonance peak values of the FRF with observation distance is shown in Figure 19. It can be observed that the resonance peak values decrease with the increase in distance between the hammering position and the acceleration measurement point. The first-order resonance peak attenuations rapidly, and it essentially disappears when the distance reaches 12 m. The attenuation of the second-order resonance peak is relatively slower, and it has not completely disappeared even when the distance reaches 40 m.

The continuous wavelet transform of the signal results in the time–frequency characteristics plot of the rail response when the hammering point is located at the sensor deployment cross-section, as shown in Figure 20. The vibration near the frequency of 1028 Hz, associated with pinned-pinned vibration, experiences a rapid attenuation in amplitude in the initial 0.01 s of vibration, followed by a duration of approximately 0.05 s. On the other hand, the vibration at 195 Hz not only experiences attenuations rapidly with distance but also has a shorter duration compared to the 1028 Hz vibration, lasting for approximately 0.02 s.

From the time-domain analysis results, it is evident that the vibration of the rail essentially ceases 0.02 s after excitation. Therefore, when applying the moving window method to process rail data, the truncation time should be set to more than 0.02 s.

When performing vehicle–track coupled dynamic calculations, it is necessary to consider the high-frequency vibrations of the rail. At a distance of 40 m, the rail vibration at 1028 Hz is still ongoing. In using the modal superposition method to calculate rail responses, since the degrees of freedom are fewer, it does not consume a significant amount of computational resources.

Additionally, due to the long attenuation distance of rail vibrations, the truncation distance can be set as large as possible when using the moving window method.

### 6.3. Slab Vibration Characteristics

Figure 21 presents the acceleration response of the slab when subjected to impact loads at different positions. From the graph, it can be observed that when the hammering point is located at the testing cross-section, the slab exhibits the maximum vibration response, with a maximum acceleration of approximately 0.20 g. Overall, as the hammering position gradually moves away from the acceleration measurement point, the initial response time of the slab gradually lags behind, and the maximum vibration acceleration of the slab also decreases accordingly. The vibration of the slab gradually attenuates over time, with the presence of new excitation peaks during the attenuation process. This is because as the vibration propagates along the rail, other clip positions on the slab are subjected to secondary excitation from the clip force. The overall response essentially ends after approximately 0.03 s.

The $H_{1}$ estimate of the slab vibration frequency response function (FRF) is shown in Figure 22. From the graph, it can be observed that peak values appear at 70 Hz, 195 Hz, and 1146 Hz, as well as at the harmonics of 50 Hz. As discussed in the subsequent time–frequency analysis, the peak at 50 Hz is attributed to interference from AC electrical signals affecting the sensor, resulting in noise with signal intensity that does not attenuation over time. Peaks appearing at the harmonics of 50 Hz are also a result of this interference.

The peaks observed at 70 Hz, 195 Hz, and 1146 Hz represent resonance peaks in response to the hammering on the slab. The variation of resonance peak values with observation distance is illustrated in Figure 23. As the hammering position gradually moves away from the acceleration measurement point, the resonance peak values decrease.

The resonance peak at 70 Hz corresponds to the natural frequency of the slab. This is demonstrated by performing a modal analysis of the finite element model of the track structure at the test section. Figure 24 shows the first three modes obtained from the modal analysis of the finite element model of the track slab. The corresponding natural frequencies are 70.35 Hz, 71.44 Hz, and 73.91 Hz, all concentrated around 70 Hz.

The resonance peak at 195 Hz attenuates significantly within the 0 to 3 m range, slows down its attenuation between 3 and 9 m, and remains generally stable from 9 to 24 m, where it finally dissipates almost entirely at 24 m.

Unlike the rail, the resonance peak at 1146 Hz attenuates rapidly on the slab. It significantly diminishes within the 0 to 3 m range and, although still attenuating, becomes generally stable between 3 and 12 m. The vibration becomes almost stable at a distance of 12 m from the excitation point.

The time–frequency characteristics plot of the slab is shown in Figure 25. Vibrations near the frequency of 1146 Hz dissipate rapidly within 0.01 s after the onset of vibration. Vibrations near the frequencies of 70 Hz and 195 Hz have durations of approximately 0.03 s.

After excitation, the vibration duration of the slab is approximately 0.03 s. However, it is important to note that this is under the continuous influence of the rail’s vibration. In other words, the intrinsic vibration duration of the slab should be the duration after the rail ceases to excite the slab through the fasteners. Therefore, the vibration duration of the slab should be around 0.01 s. Consequently, when applying the moving window method in vehicle–track coupled dynamic calculations, the truncation time of the time window should be set to more than 0.01 s to accurately reflect the vibration response of the track structure.

The vibrations of the slab and tunnel wall are generally considered to be below 200 Hz. For the slab, the vibration attenuates rapidly within the 0 to 9 m range, meaning that the truncation distance of the moving window should be at least 9 m to accurately reflect the vibration of the slab. If higher accuracy is required, the truncation distance should be set to at least 32 m.

### 6.4. Tunnel Vibration Characteristics

Figure 26 illustrates the acceleration response of the tunnel when subjected to impact loads at different positions. From the graph, it can be observed that when the hammering point is located at the testing cross-section, the tunnel exhibits the maximum vibration response, with a maximum acceleration of approximately 0.10 g. As the hammering position gradually moves away from the acceleration measurement point, the initial response time of the tunnel gradually lags behind, and the maximum vibration acceleration of the tunnel also decreases accordingly. Similar to the slab, the vibration of the tunnel gradually attenuates over time, with the presence of new excitation peaks during the attenuation process. The overall response essentially ends after approximately 0.03 s.

The $H_{1}$ estimate of the tunnel vibration frequency response function (FRF) is depicted in Figure 27. From the graph, it can be observed that resonance peaks of the tunnel occur at positions around 195 Hz and 1146 Hz.

The variation of resonance peak values with observation distance is illustrated in Figure 28. As the hammering position gradually moves away from the acceleration measurement point, the resonance peak values rapidly decrease. The resonance peak at 195 Hz attenuations slower with distance compared to the slab, essentially disappearing when the distance from the excitation point reaches 32 m. In contrast, the resonance peak at 1146 Hz attenuations more rapidly on the tunnel compared to the slab, essentially disappearing when the distance from the excitation point reaches 10 m.

Comparing with the FRF of the slab, it can be observed that during the transmission of rail vibrations to the tunnel, vibrations below 195 Hz are essentially attenuated. The track arch plays a significant role in suppressing low-frequency vibrations. Vibrations around 195 Hz experience approximately a halved amplitude attenuation during transmission, but they are more difficult to dissipate within the tunnel. On the other hand, high-frequency vibrations at 1146 Hz experience almost no amplitude attenuation during transmission from the slab to the tunnel, but they dissipate more rapidly within the tunnel.

The time–frequency characteristics of the tunnel are depicted in Figure 29. Similar to the slab, vibrations near 1146 Hz dissipate rapidly within 0.01 s after the onset of vibration. Vibrations near 195 Hz have a duration of approximately 0.03 s. However, vibrations near 70 Hz fail to propagate through the track arch to the tunnel.

Similar to the slab, the truncation time for the tunnel should also be at least 0.01 s. For the truncation distance, the vibration of the tunnel attenuates rapidly within the 0 to 12 m range. Therefore, the truncation distance of the moving window should be at least 12 m to accurately reflect the vibration of the tunnel. If higher accuracy is required, the truncation distance should be set to 32 m.

### 6.5. Vibration Transmission Characteristics Between Track Structures

By examining the frequency domain characteristics of different structural vibration signals, it can be observed that both the rail, slab, and tunnel exhibit resonance peaks at around 195 Hz. Another resonance peak frequency is observed, with the rail at approximately 1028 Hz and the slab and tunnel around 1146 Hz. Due to the presence of nonlinear materials such as rubber pads within the fastening system, it can be inferred that the vibration at 1146 Hz in the slab and tunnel is transmitted from the rail’s vibration at 1028 Hz, with this frequency change attributed to the nonlinearity of the fastening system. Resonance peaks at around 70 Hz are exclusively present in the slab structure, likely due to the inherent frequency of the slab.

To analyze the transmission characteristics of vibrations at different frequencies among the track structures, the peak FRF values of the vibration responses at resonance peak frequencies are extracted for comparison when the hammering point is at 1st position. The results are summarized in Table 1.

The data in the table indicate that the 195 Hz vibration attenuates rapidly when transmitted from the rail to the slab, with the FRF magnitude decreasing by approximately 98.62%. However, the attenuation is less pronounced when the vibration is transmitted from the slab to the tunnel, with a reduction in magnitude of approximately 73.33%.

For the 1028 Hz and 1146 Hz vibrations transmitted from the rail to the slab, a similar rapid attenuation is observed, with the FRF magnitude decreasing by approximately 99.08%. However, when transmitted from the slab to the tunnel, the attenuation is less significant, with a reduction in magnitude of only about 55.56%.

This demonstrates that for the continuous slab, vibrations attenuate rapidly when transmitted between structures. The fastening system exhibits a high isolation effect for both the 195 Hz, 1028 Hz, and 1146 Hz vibrations. When the slab vibration is transmitted through the arch to the tunnel, the arch demonstrates a higher isolation effect for the 195 Hz vibration compared to the 1028 Hz and 1146 Hz vibrations.

## 7. Conclusions

Rail hammering experiments and train-induced vibration response tests were conducted in this paper. By analyzing the vibration response of the track structures caused by the impact, the main conclusions obtained are as follows.

(1) The resonance peaks of the rail response under hammer excitation are located at 195 Hz and 1028 Hz. Vibrations at 195 Hz attenuate rapidly with distance, essentially disappearing when the distance reaches 12 m. The attenuation over time is similarly rapid, with a duration of only approximately 0.02 s. Vibrations at 1028 Hz attenuation more slowly with distance; the vibrations have not fully dissipated, even when the distance reaches 40 m. The attenuation over time lasts approximately 0.01 s.

(2) Under hammer excitation, the resonance peaks of the slab response are located at 70 Hz, 195 Hz, and 1146 Hz. The 70 Hz frequency corresponds to the first natural frequency of the slab. Vibrations at this frequency attenuate rapidly with distance, particularly in the range of 0–3 m, with a duration of approximately 0.03 s. Vibrations at 195 Hz also attenuate rapidly with distance, and essentially disappear when the distance reaches 24 m, with a duration of 0.03 s. Vibrations at 1146 Hz attenuation more slowly with distance, and essentially disappear when the distance reaches 12 m, with a duration of only 0.01 s.

(3) Under hammer excitation, the resonance peaks of the tunnel response are located at 195 Hz and 1146 Hz. Vibrations at 195 Hz attenuation slowly with distance, and they essentially disappear when the distance reaches 32 m. The duration of the vibrations at this frequency is also 0.03 s. Vibrations at 1146 Hz attenuate rapidly with distance, and essentially disappear when the distance reaches 10 m, in comparison to the slab. The duration of vibrations at this frequency is only 0.01 s.

(4) From the response amplitudes of FRF, it is evident that vibrations exhibit significant attenuation as they propagate from the rail to the slab and then to the tunnel. For vibrations at frequencies of 195 Hz, 1028 Hz, and 1146 Hz, the FRF amplitudes decreased by 98.62% and 99.80%, respectively, when transmitted from the rail to the slab. The FRF amplitudes decreased by 73.33% and 55.56%, respectively, when transmitted from the slab to the tunnel.

(5) When performing vehicle–track coupled dynamic calculations, the truncation time for the moving window should be at least 0.02 s for the rail, with a truncation distance greater than 40 m. For the slab and tunnel, the truncation time should be at least 0.01 s. The truncation distance for the slab should be at least 9 m, and for the tunnel, it should be at least 12 m.

(6) By comparing the vibration response induced by the train with the hammering response of the rail, it is found that the main excitation frequencies of the subway type B car are concentrated around 400 Hz, 600 Hz, and 900 Hz.

Additionally, the findings on vibration frequency characteristics and the transmission characteristics between structures reveal the inherent properties of the track structure. These insights are valuable for designing vibration mitigation measures, allowing for the development of targeted isolation facilities to address vibrations of different frequencies in various structures.

## Figures and Tables

**Figure 1 sensors-25-00868-f001:**
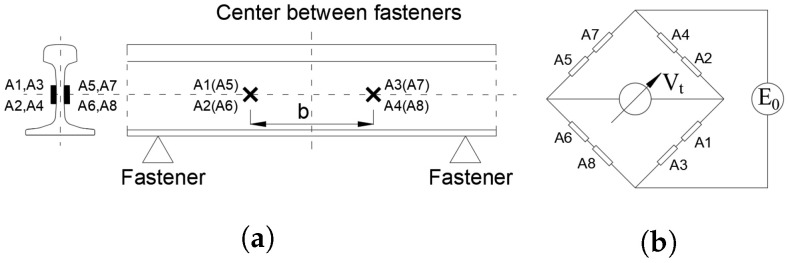
Vertical wheel–rail force testing device. (**a**) Arrangement of strain gauge rosettes. (**b**) Strain gauge bridge connection method.

**Figure 2 sensors-25-00868-f002:**
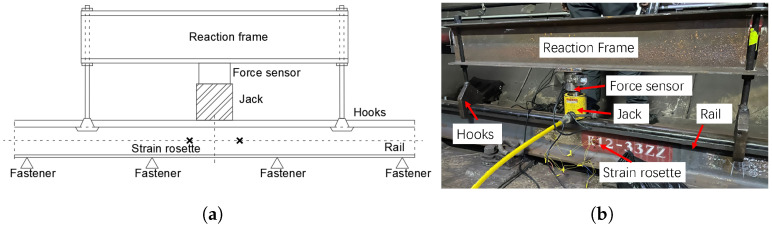
Calibration device for the wheel–rail force. (**a**) Schematic diagram of the vertical calibration device. (**b**) Calibration device for vertical force.

**Figure 3 sensors-25-00868-f003:**
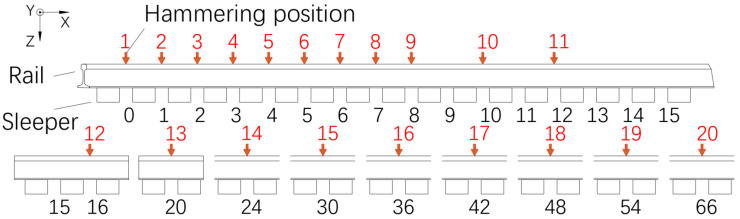
Schematic diagram of the hammering point location.

**Figure 4 sensors-25-00868-f004:**
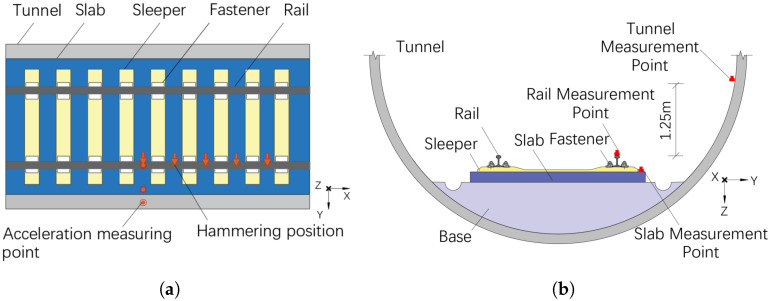
The acceleration measuring point. (**a**) Top view. (**b**) Cross-section view.

**Figure 5 sensors-25-00868-f005:**
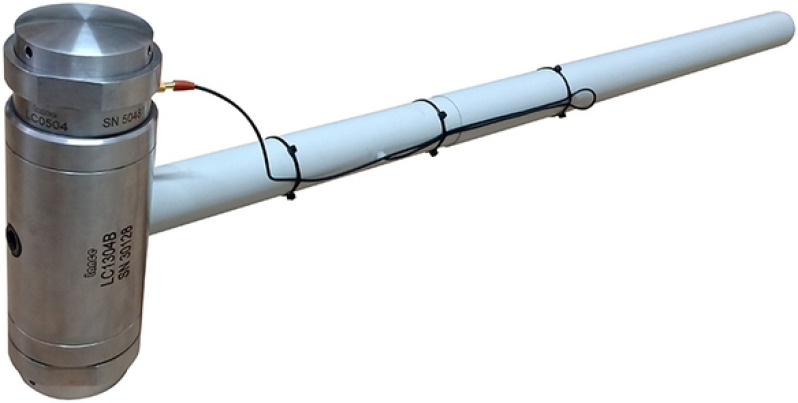
Hammer strike excitation equipment.

**Figure 6 sensors-25-00868-f006:**
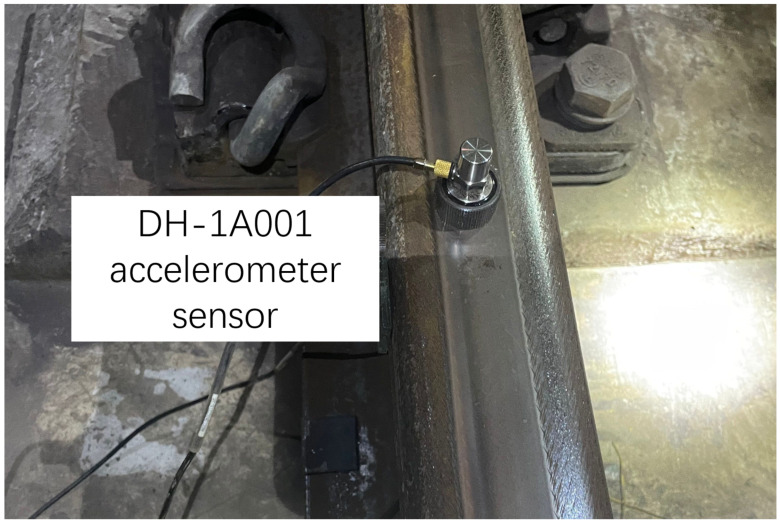
Accelerometer sensor for the rail.

**Figure 7 sensors-25-00868-f007:**
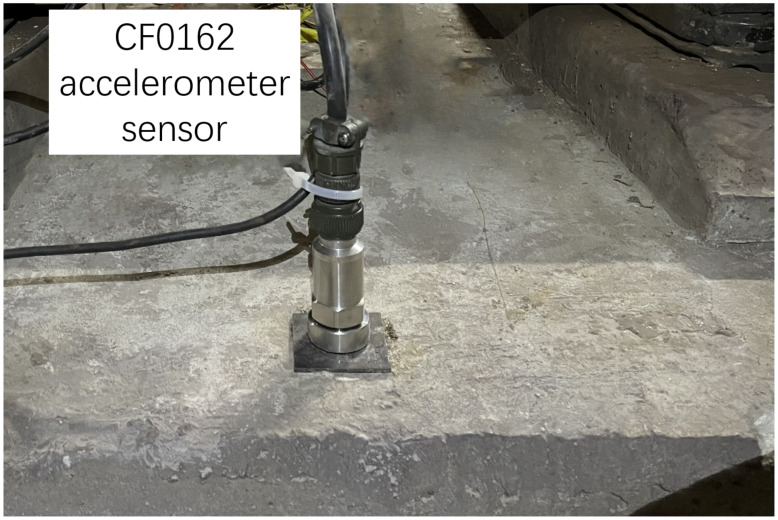
Accelerometer sensor for the slab.

**Figure 8 sensors-25-00868-f008:**
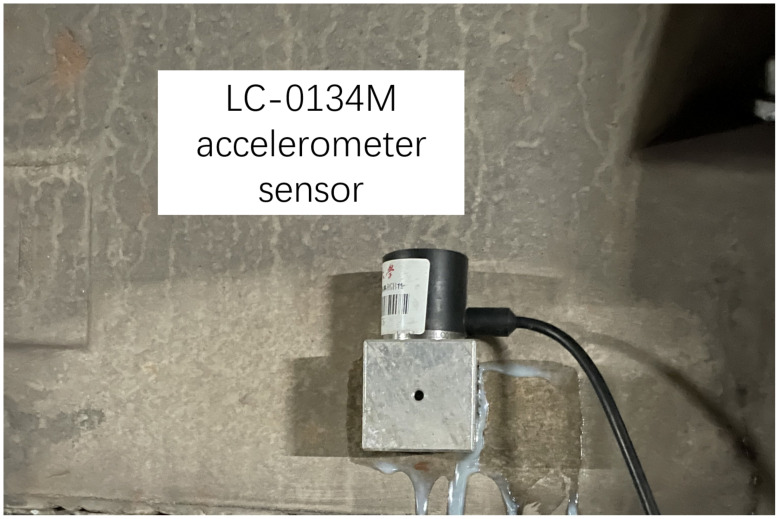
Accelerometer sensor for the tunnel.

**Figure 9 sensors-25-00868-f009:**
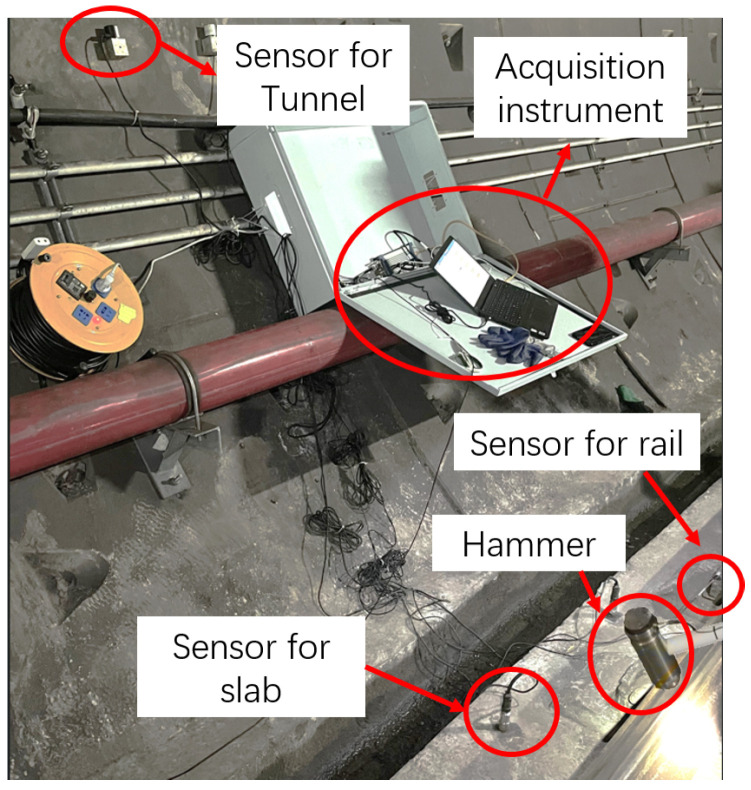
Accelerometer sensor for the tunnel.

**Figure 10 sensors-25-00868-f010:**
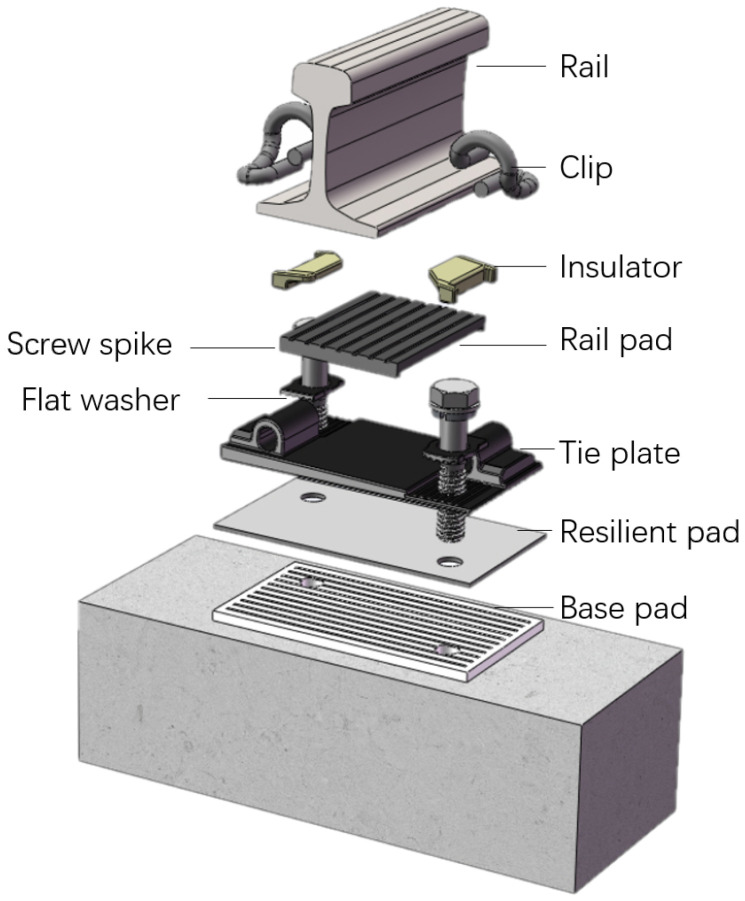
Fastener.

**Figure 11 sensors-25-00868-f011:**
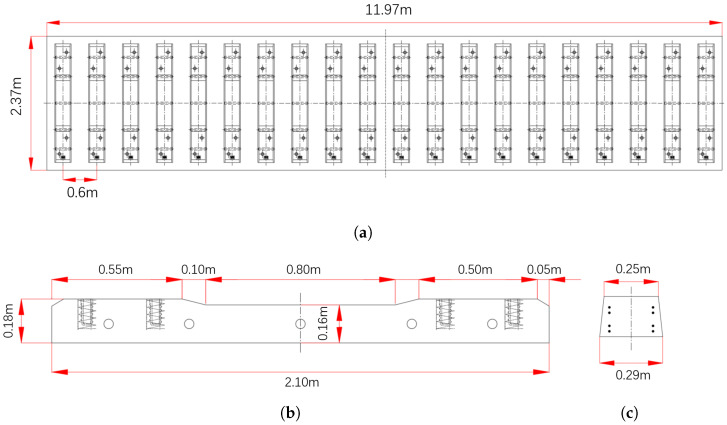
Slab and sleeper. (**a**) Slab. (**b**) Front view of the sleeper. (**c**) Side view of the sleeper.

**Figure 12 sensors-25-00868-f012:**
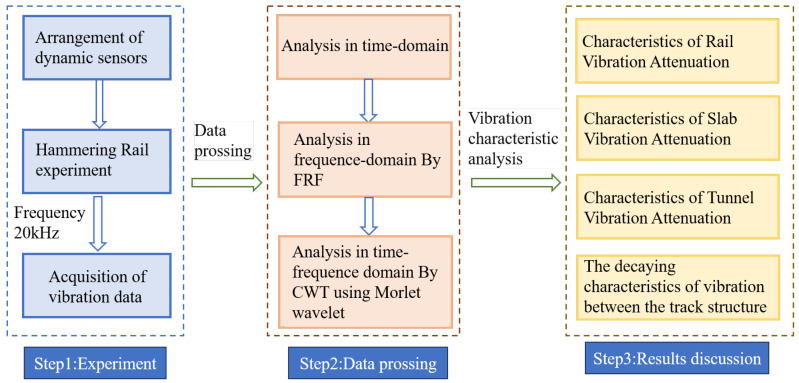
Workflow.

**Figure 13 sensors-25-00868-f013:**
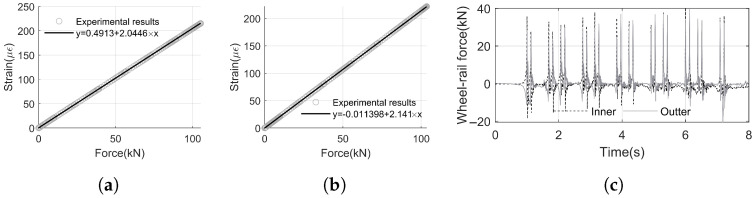
Vertical wheel–rail force. (**a**) Calibration of the wheel–rail force (inner rail). (**b**) Calibration of the wheel–rail force (outer rail). (**c**) Wheel–rail force.

**Figure 14 sensors-25-00868-f014:**
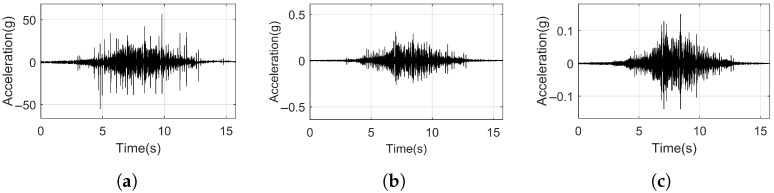
Time history vibration. (**a**) Rail. (**b**) Slab). (**c**) Tunnel.

**Figure 15 sensors-25-00868-f015:**
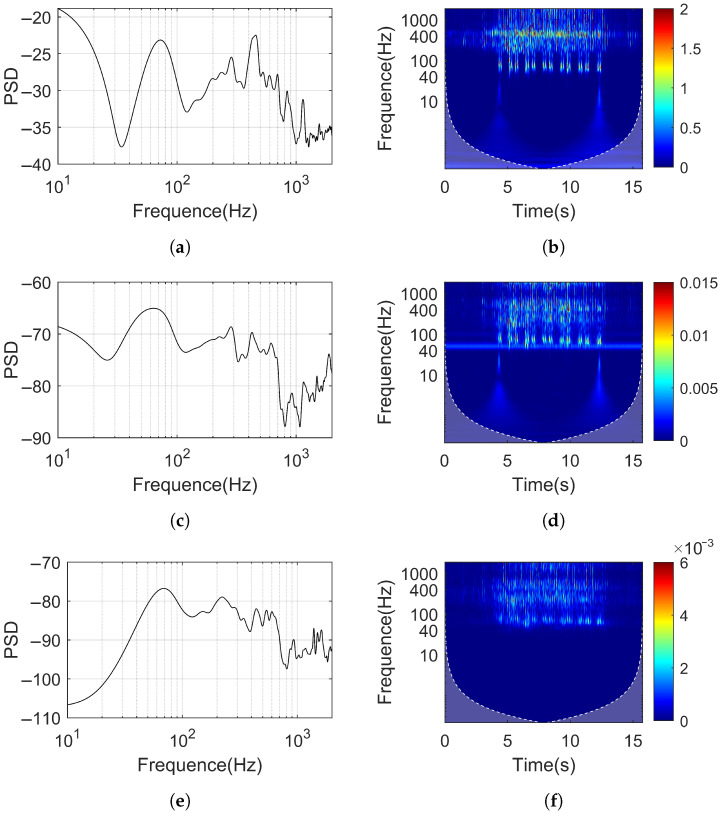
Power spectral density and time–frequency diagram. (**a**) PSD of the rail. (**b**) Time–frequency diagram of the rail). (**c**) PSD of the slab. (**d**) Time–frequency diagram of the slab. (**e**) PSD of the tunnel. (**f**) Time–frequency diagram.

**Figure 16 sensors-25-00868-f016:**
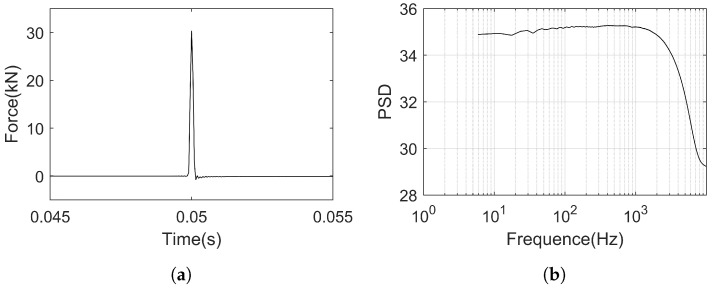
Hammer impact signal. (**a**) Time history curve. (**b**) Power spectral density (PSD).

**Figure 17 sensors-25-00868-f017:**
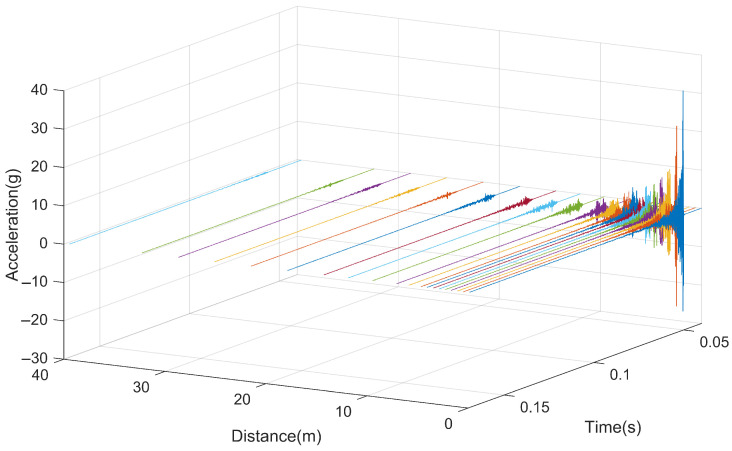
Time history curve of rail vibration.

**Figure 18 sensors-25-00868-f018:**
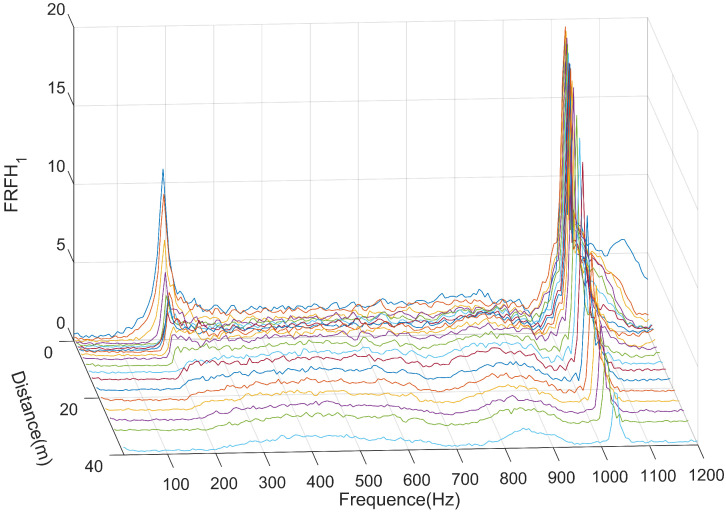
The FRF of rail vibration.

**Figure 19 sensors-25-00868-f019:**
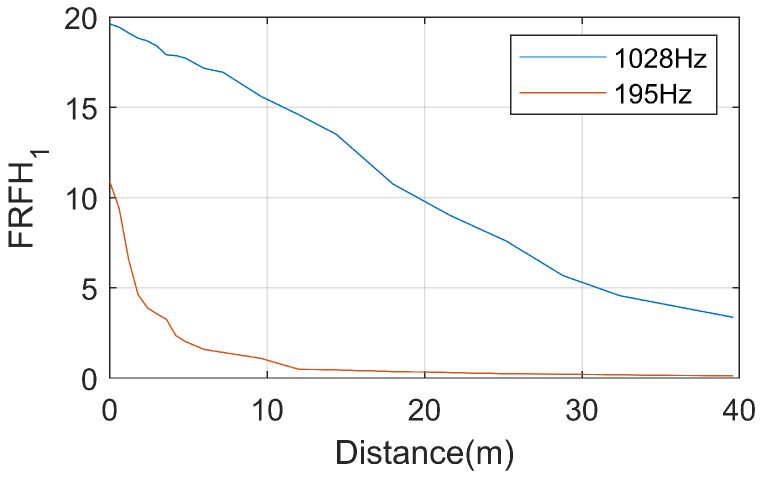
The peak value of rail resonance attenuations with distance.

**Figure 20 sensors-25-00868-f020:**
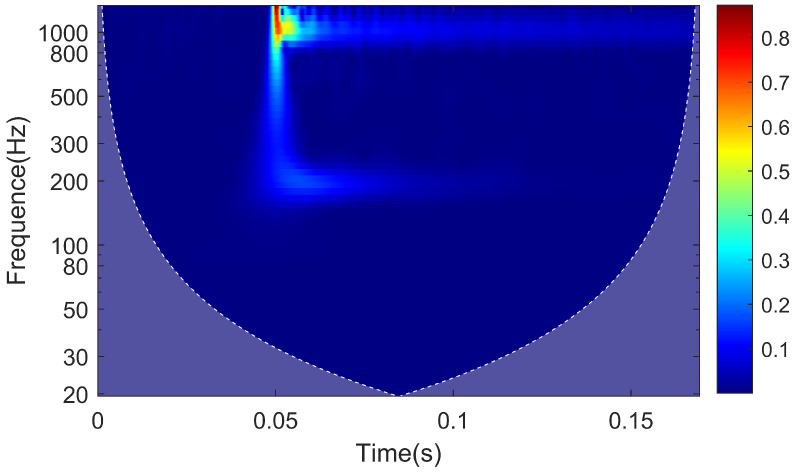
Time–frequency characteristic map of rail response.

**Figure 21 sensors-25-00868-f021:**
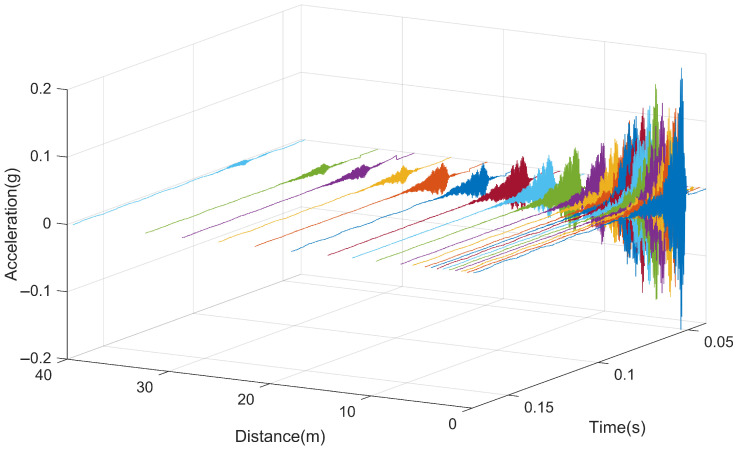
Time history curve of the slab vibration.

**Figure 22 sensors-25-00868-f022:**
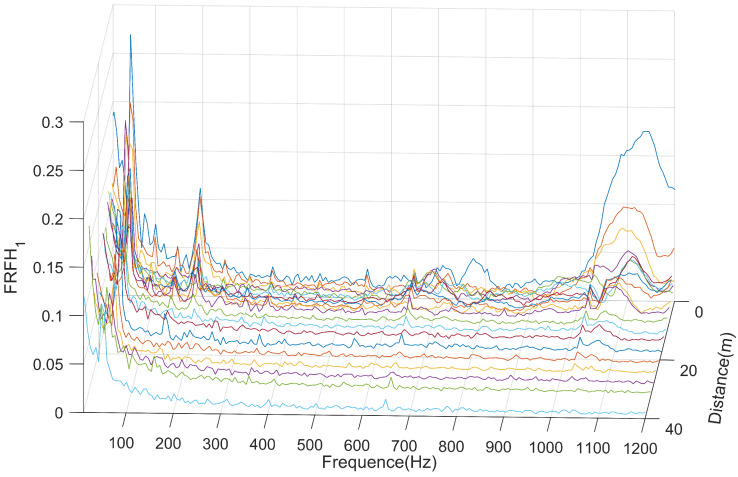
The FRF of the slab vibration.

**Figure 23 sensors-25-00868-f023:**
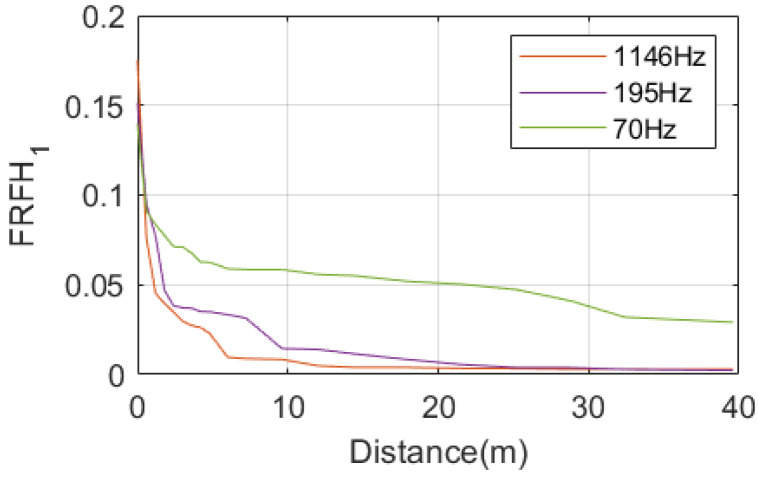
The peak value of slab resonance attenuations with distance.

**Figure 24 sensors-25-00868-f024:**
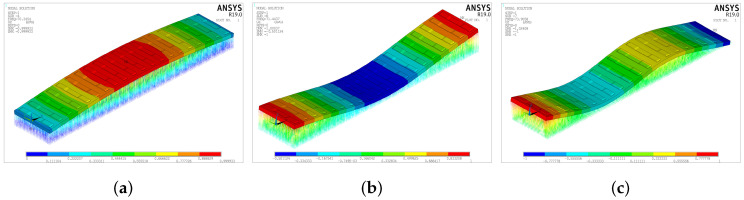
Model of the slab. (**a**) First order. (**b**) Second order. (**c**) Third order.

**Figure 25 sensors-25-00868-f025:**
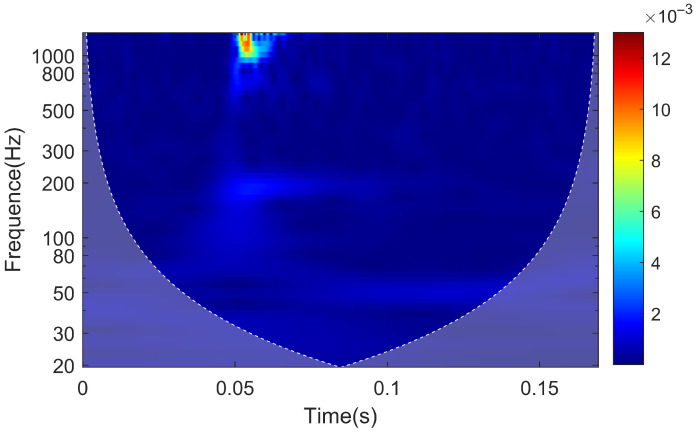
Time–frequency characteristic map of the slab response.

**Figure 26 sensors-25-00868-f026:**
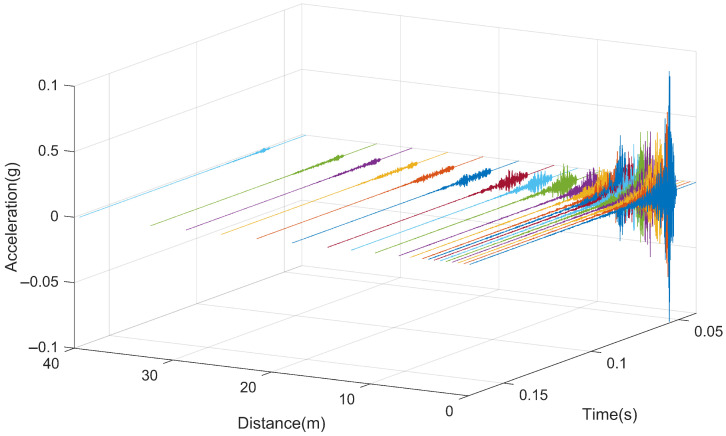
Time history curve of the tunnel vibration.

**Figure 27 sensors-25-00868-f027:**
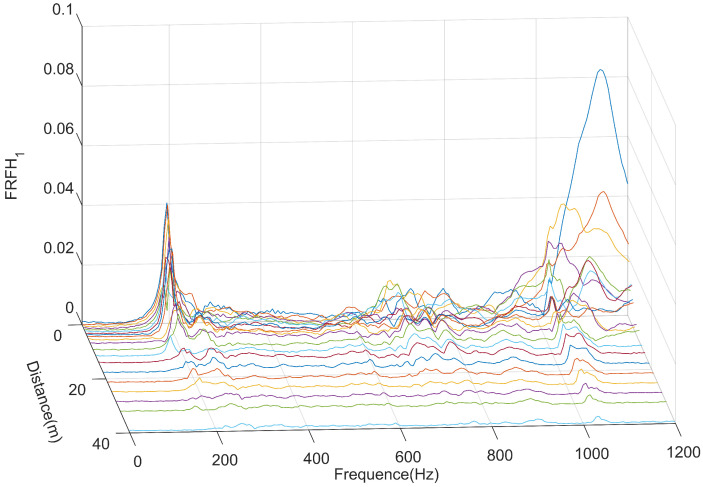
The FRF of the tunnel vibration.

**Figure 28 sensors-25-00868-f028:**
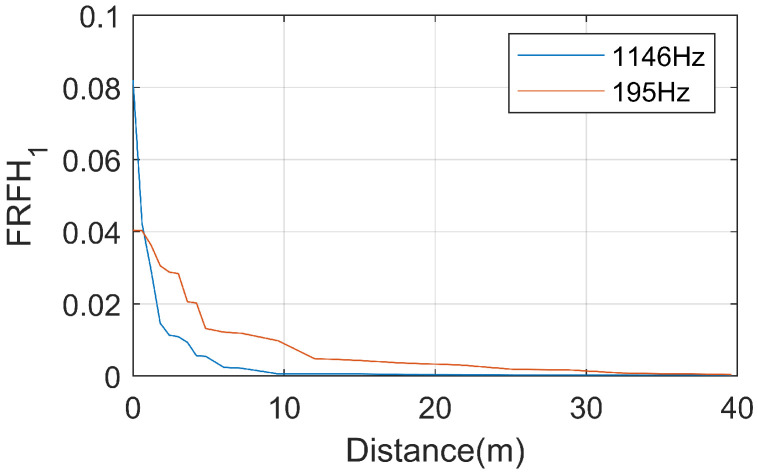
The peak value of tunnel resonance attenuations with distance.

**Figure 29 sensors-25-00868-f029:**
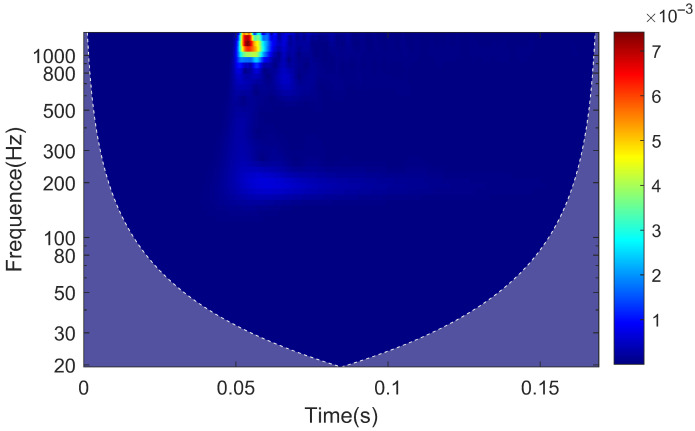
Time–frequency characteristic map of the tunnel response.

**Table 1 sensors-25-00868-t001:** Attenuation of vibrations at different frequencies between track structures.

Structure	Rail	Slab	Tunnel
195 Hz	10.90	0.15	0.04
1028 Hz and 1146 Hz	19.63	0.18	0.08

## Data Availability

The authors are unable or have chosen not to specify which data have been used.

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
