# Peer review of "Experimental Study on Vibration Attenuation Characteristics of Ballastless Track Structures in Urban Rail Transit"

_sensors, 2025, doi:10.3390/s25030868_

Round 1
Reviewer 1 Report
Comments and Suggestions for Authors
The vibration characteristics and attenuation laws of track structures are crucial for the design and simulation calculations of track systems, and have long been a focal point of scholarly research. This article conducts a hammering test on the rail within a tunnel, providing an in-depth analysis of the vibration characteristics of the track structure from multiple dimensions, revealing the propagation and attenuation patterns of these vibrations. The study designs a novel experimental method, determining the magnitude of the impact force based on measured wheel-rail forces. Accelerometers are placed on the rail, slab, and tunnel wall to record vibration responses at different hammering positions. The analysis covers various dimensions including time domain, frequency domain, time-frequency domain, and spatial aspects, highlighting the attenuation laws of vibrations of different frequencies across time, space, and the track structure. The overall structure of the paper is logical and coherent, employing reliable analytical methods to reach novel conclusions, thereby exhibiting a degree of innovation. It generally meets the publication requirements for this journal and is recommended for publication after necessary modifications.
(1) In the experimental methods section, the paper provides information about the measurement equipment used for collecting track structure vibration data, including model, range, and sampling frequency. However, while the method for testing wheel-rail forces is described, the specific model of the strain gauge rosettes used and the sampling frequency are not mentioned. It is recommended to include this information.
(2) There are a few spelling errors in the text; for example, in the abstract, “Than the Lance-LC1304B force hammer was...” should be “Then the Lance-LC1304B force...”. On page 5, lines 188 and 189, there are formatting errors where there is a lack of space between numbers and units. The conclusion section repeatedly uses “attenuation rapidly (or slowly),” which should be changed to “attenuate rapidly (or slowly).” In the conclusion section on page 19, line 537, “For vibrations at frequencies of 195 Hz and 1028 Hz & 1146 Hz,” should be revised to “For vibrations at frequencies of 195 Hz, 1028 Hz, and 1146 Hz.”. The authors are advised to carefully review and correct these errors.

Reviewer 2 Report
Comments and Suggestions for Authors
The paper presents a detailed study of an impulse response of the subway track structures. Although the paper is detailed, well written and presents rather interesting results, we have some minor concerns. On several occasions, the authors provide definitions that may be considered a common knowledge (at least by the readers of this journal) -- for instance, in lines 272 -- 277 the definition of the frequency response function is given. We feel that by omitting such well-known facts the paper will be shorter and more concise. Apart from this, we feel that the results presented in the paper present a potential valuable resource to any research dealing with railway structures dynamics.
Reviewer 3 Report
Comments and Suggestions for Authors
Review comments:
(1) The article starts from the experiment and summarizes the corresponding experimental phenomena. It is suggested that the authors add the theoretical equations for the propagation of shock excitations in orbits, thus verifying the credibility of the experiments.
(2) The article distinguishes different location impacts based on the fluctuation characteristics of the response spectrum. The author needs to supplement theoretical analysis on the characteristic frequency bands of impacts at different positions, and whether they match the experimental frequency bands?
(3) According to experience, the impact characteristics at different positions are generally unstable, and the cloud images obtained from time-frequency analysis cannot obtain stable grayscale differentiation. Suggest the author to use large sample analysis for impact experiments at different locations to avoid idealized conclusions caused by individual experimental samples.
(4) The impact and propagation processes of the orbit are both time-varying, and the method used in the article corresponds to stabilizing the signal. Suggest the author to explain whether the actual time-varying signal is regular and stable, and how to handle time-varying signals.
(5) The actual orbital propagation signals are coupled signals or even strong background noise signals, and even experimental signals have interference from rigid vibrations between components. The entire article is actually the process of extracting effective features from response signals. Therefore, the author needs to supplement the preprocessing methods or noise reduction techniques for experimental signals.
